# Large-Scale Trade in a Songbird That Is Extinct in the Wild

Vincent Nijman [1,*], Marco Campera [1,2,*], Ahmad Ardiansyah [2,3], Michela Balestri [1,2], Hani R. El Bizri [1], Budiadi Budiadi [4], Tungga Dewi [2], Katherine Hedger [2], Rifqi Hendrik [2], Muhammad Ali Imron [4], Abdullah Langgeng [2,5], Thais Q. Morcatty [1], Ariana V. Weldon [1] and K. A. I. Nekaris [1,2]

1   Oxford Wildlife Trade Research Group, Oxford Brookes University, Oxford OX3 0BP, UK; mbalestri@brookes.ac.uk (M.B.); hel-bizri@brookes.ac.uk (H.R.E.B.); tqueiroz-morcatty@brookes.ac.uk (T.Q.M.); a.weldon@brookes.ac.uk (A.V.W.); anekaris@brookes.ac.uk (K.A.I.N.)
2   Little Fireface Project, Cipaganti, Garut 44163, Indonesia; ahmadardiansy@gmail.com (A.A.); tdhp25@gmail.com (T.D.); katey.hedger@gmail.com (K.H.); rifqihendrik@gmail.com (R.H.); abdullahlanggeng@gmail.com (A.L.)
3   Forest and Nature Conservation Policy Group, Wageningen University, 6708 PB Wageningen, The Netherlands
4   Faculty of Forestry, Universitas Gajah Madah, Yogyakarta 55281, Indonesia; budiadi@ugm.ac.id (B.B.); maimron@ugm.ac.id (M.A.I.)
5   Primate Research Institute, Kyoto University, Kyoto 484-8506, Japan
*   Correspondence: vnijman@brookes.ac.uk (V.N.); mcampera@brookes.ac.uk (M.C.)

**Abstract:** Indonesia is at the epicenter of the Asian Songbird Crisis, i.e., the recognition that the cage bird trade has a devastating impact on numerous imperiled bird species in Asia. The Javan pied starling *Gracupica jalla*, only in the last five years recognized as distinct from the pied starlings of mainland Southeast Asia, has been declared extinct the wild in 2021. Up until the 1980s, it used to be one of the most common open countryside birds on the islands of Java and Bali, Indonesia. From the early 2000s onwards, the species is commercially bred to meet the demand from the domestic cagebird trade. We conducted 280 market surveys in 25 bird markets in Java and Bali between April 2014 and March 2020, with 15 markets being surveyed at least six times. We recorded 24,358 Javan pied starlings, making it one of the most commonly observed birds in the markets. We established that, conservatively, around 40% of the birds in the market were sold within one week and used this to estimate that at a minimum ~80,000 Javan pied starlings are sold in the bird markets on Java and Bali. The latter represents a monetary value of USD5.2 million. We showed that prices were low in the 1980s, when all birds were sourced from the wild. It became more varied and differentiated in the 2000s when a combination of now expensive wild-caught and cheaper captive-bred birds were offered for sale, and prices stabilized in the 2010s when most, if not all birds were commercially captive-bred. Javan pied starlings are not protected under Indonesian law, and there are no linked-up conservation efforts in place to re-establish a wild population on the islands, although small-scale releases do take place.

**Keywords:** Asian Songbird Crisis; conservation evidence; Indonesia; Javan pied starling; *Sturnus contra*; wildlife trade

## 1. Introduction

The Indonesian island of Java and, to a lesser degree, neighboring Bali are at the epicenter of the Asian Songbird Crisis, i.e., the recognition that the ongoing cage bird trade in various parts of Asia has a devastating impact on numerous imperilled bird species [1]. Birds are kept as companion animals for their varied plumage or singing abilities, to compete in singing competitions, to be used as "masters" whereby their songs improve the repertoire of the songs of other songbirds that do enter singing competitions, or to be shown as a status symbol. Birds are trapped in the thousands both by professional and

part-time bird catchers, including farmers, villagers, and city folk, brought to middlemen and transported to numerous bird markets that are found in just about every major city on the islands [2]. The capital Jakarta alone has at least four dedicated bird markets, and in the largest of these, Pramuka, Chng et al. [3] recorded 16,171 largely wild-caught birds of 190 species during a single visit in 2015. Comparable numbers for this market were reported in the late 1980s [4] and the early 1990s [5]. This persistent and large-scale threat has had a destructive effect on numerous species, both ones that are only found on (parts of) Java and ones that range over larger parts of Southeast Asia [3,5–12].

Baveja et al. [13] and van Balen and Collar [14] set out the case that the Javan pied starling *Gracupica jalla* is a distinct species that is extinct in the wild. Baveja et al. [13], using historic and contemporary genetic samples, provided support for the recognition of *G. jalla* from Java and Bali as distinct from other pied starling species from mainland Asia (*G. floweri*, *G. contra*, *G. supercilliaris*: pied starlings do not occur on the Thai-Malay Peninsula, Borneo, and most of Sumatra, and *G. jalla* is geographically isolated). Previously considered part of a wide ranged Asian pied starling complex, in 2016 its IUCN threat status was assessed as Critically Endangered [15]. Van Balen and Collar [14] lamented that "For a once-common bird to disappear completely from an area the size of Java and Bali (134,000 km$^2$) without anyone raising the alarm is unique in avian conservation. The circumstance is also unique because there is no other case on earth in which a bird species has disappeared from the wild and yet is retained in captivity by commercial interests which have zero engagement with conservation agencies."

Even though trade is considered one of the main reasons for bird species' decline, we have surprisingly little data on their numbers in trade. What we do have originates from one-off visits to a small number of bird markets with no longitudinal data. Van Balen and Collar [14] summarized what was known on the basis of these short visits by four research teams and reported it as common in one bird market in 1980 (up to 7% of the birds present in the market), and less common in one bird market in 1989, three bird markets in 2008, and eight bird markets in 2014–2015 (all less than 4% of all birds present in the market). This decline is thought to be due to a decrease in the number of Javan pied starlings in trade rather than an increase in the total number of birds in these markets. They further noted that in recent years, most of the pied starlings in trade were either captive-bred Javan pied starlings or possibly imported Asian pied starlings [14].

We monitored the trade in Javan pied starling since 2014 on the islands of Java and Bali. Here, we present data that fill an important gap in our knowledge about how the cage bird trade could lead to the extirpation of this species that was one of the most common birds of the Javan countryside. By focusing on a large number of bird markets, all surveyed at least twice, we expected to show that the species is traded in even more substantial numbers than previously thought and highlight the role commercial captive breeding plays in the trade. This will also give us an opportunity to comment on the conservation needs and the conservation value of Javan pied starlings in trade.

## 2. Materials and Methods

### 2.1. Data Acquisition—Market Surveys

We conducted 280 market surveys of 25 bird markets in 16 cities in Java and Bali between April 2014 and February 2020 (Figure 1). Listed from west to east these were: Pramuka, Jatinegara, Barito, Kebayoran Lama, Cipinang, all in the Jakarta Capital Region; Pelongan (in the city of Depok), Rawa Lumbu (Bekasi), Empang (Bogor), Pasundan (Sukabumi), Pasirhayan (Cianjur), Sukahaji (Bandung), Bayongbong, Kerkhof and Mawar (Garut), Cikurubuk (Tasikmalaya), Plered (Cirebon), all in the province of West Java; Karimata (Semarang) and Depok (Surakarta) in the province of Central Java; Pasty in the Yogyakarta Special Region; Bratang and Turi (Surabaya), Splindit (Malang) in the province of East Java; Bringkit (Mengwi), Sanglah, and Satria (Denpasar) on Bali. Each market was surveyed two times or more, at least one month apart (we refer to these as monthly surveys), for a mean number of 11 monthly surveys per bird market, ranging from 2 to

38. For a survey, the market was visited by one or two of the authors by slowly walking through the market; birds (and other animals) that were of interest were counted. This information was entered into a mobile phone in the market or recorded in a notebook after leaving the market. Trade in the bird markets is open, and there is no need to resort to undercover techniques. Javan pied starlings are displayed in cages in front of the shop or inside shops (Figure 1); as the aim of the trader is to sell the birds to potential customers, there is no reason for them to keep them hidden in the back of their shops. The bird markets range in size from one with a dozen or so small stalls that offer a few hundred of birds for sale to four story buildings with 200+ stalls and shops that offer wild-caught birds, captive-bred birds, cages, bird food, and other pet supplies.

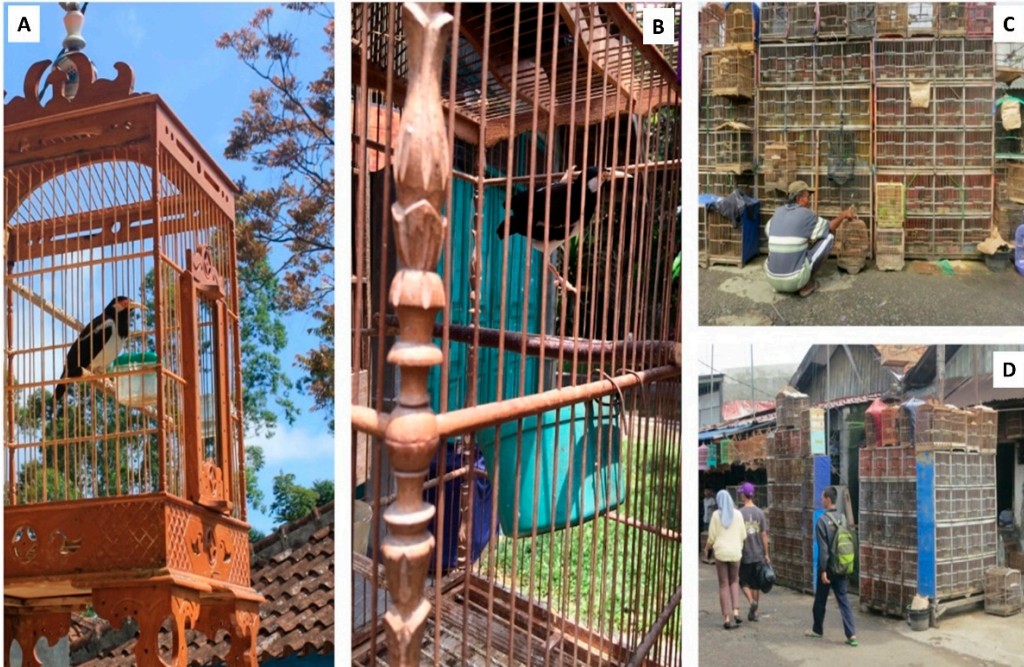

**Figure 1.** Javan pied starling in captivity in Java, Indonesia, in a singing competition (**A**) and for sale in Bandung (**B**), with representation of the conditions in two bird markets—Sukahaji, Bandung (**C**) and Jatinegara, Jakarta (**D**).

We revisited Kerkhof bird market in Garut, West Java more frequently at shorter intervals throughout the year to estimate turnover. There were 8 instances whereby the number of days between the first survey and the second survey was five days; we revisited the market 28 times when there were seven days between the first and the second survey. There were five instances where there were nine days between the first and the second survey and 14 when there were 14 days between the first and the second survey.

Two of the authors, Vincent Nijman (VN) and Mohammed Ali Imron (MAI) have observed the trade in Javan pied starlings in the bird markets in Java recurrently from the early 1990s to 2019, with VN having a geographical bias towards Jakarta and the province of West Java, and MAI having a geographic bias towards Yogyakarta and Central Java. Network analysis and network perspectives have been suggested to offer a means to investigate the interactions that govern various aspects of the trade in songbirds [16,17]. In our study, this was partially guided by personal experience such as visiting bird markets, observing the trade in songbirds, spending time in places where cagebird enthusiasts congregate, and a review of the literature (including government documentation and legislation and reports from NGOs such as BirdLife International Indonesia Programme/Burung Indonesia, Fauna and Flora International Indonesia Programme, WWF Indonesia, and the Asian Wetland Bureau). We also obtained information on the trade from documentation and discussions on social media, including blogs and vlogs [17]. Over the course of the surveys, we held

discussions with commercial breeders of Javan pied starlings (and other species of star-lings) about their business. This included visits to the villages of Klaten in central Java and Wukirsari in Yogyakarta, widely seen as the center for the commercial breeding of starlings (e.g., [5,18]). For 2018, it was reported that there were 1031 pied starling breeders in Klaten with a stock of 20,426 birds [19]. The trade in Javan pied starling is legal in Indonesia and hence breeders and traders openly and freely shared information.

### 2.2. Data Acquisition—Monetary Value

Asking prices in the bird markets were very much dependent on the real, perceived, or potential singing abilities of the bird on offer, the age (younger birds are cheaper), sex (females are cheaper), number (a pair is cheaper per bird than single birds), and perceived or proven ability to breed. Prices appeared also much dependent on negotiation and bargaining (something we did not do). Price data were obtained from the market surveys, (online) reports, and online platforms (olx.com, topopedia.com, jualo.com, Facebook, access on 10 December 2020) on which Javan pied starlings are sold. These prices normally refer to asking prices or first quotes and may be lowered after bartering or when more than one bird is purchased. Price data from earlier years were corrected for inflation to December 2020.

### 2.3. Analysis

We used data from all 25 bird markets to establish the relationship between bird market size (i.e., the average number of stalls or shops that sell birds, scored on a 1 to 5 scale; Table 1) and the mean number of birds we encountered per survey. We used data from 15 bird markets that were surveyed at least six times to estimate levels of trade within bird markets.

**Table 1.** Proportion (mean ± s.e.m. and range) of non-*jalla* type of birds recorded in bird markets in Java and Bali while surveying Javan pied starlings.

| Bird Market Size (Classes) | N | Birds Observed | Proportion *jalla* |
|---|---|---|---|
| Small (1,2) | 4 | 105 | 98.23 ± 1.03 [96.15–100] |
| Intermediate (3,4) | 9 | 541 | 96.39 ± 1.88 [84.62–100] |
| Large (5) | 3 | 967 | 93.62 ± 4.35 [84.93–98.41] |
| All (1–5) | 16 | 1613 | 96.33 ± 1.32 [84.62–100] |

In sixteen visits, we made a more detailed morphological assessment to establish what proportion of the trade comprised *jalla*-type bird and *floweri*- or *contra*-type birds, involving a total of 1613 birds (Table 1). Morphologically these species are recognizable by the extent of bare red/orange facial skin and eye color [13].

We calculated turnover (birds being sold or having died in the markets) [20]. When revisiting a bird market at short intervals, we subtracted the number observed during the second visit from what was observed during the first visit. When the number was positive (i.e., the numbers in the market had decreased), something that was observed in 59% of the short interval surveys, these were included to estimate how many birds were sold or had died between the first and the second visit. It was not possible to calculate turnover when in between surveys a new consignment of Javan pied starlings had arrived and the number of starlings in the shops had increased. Calculating turnover in this manner only gives a minimum estimate, and turnover does not increase in a linear fashion over time ([20]; and Section 3). Turnover is expected to be high on the first few days after new birds arrive, as buyers are anticipating these new arrivals, but then it levels off. Furthermore, the longer the period between the first and second survey, the higher the likelihood that new birds had arrived and been sold without us being able to detect it, thus lowering the turnover estimate. Using the relationship between turnover and time (in days) we calculated annual numbers of turnover for each market in which we observed Javan pied starlings for sale and combined them for an overall annual estimate.

For calculating the monetary value of the total number of birds that were sold, we used a reasonable mid-point estimate of USD65 per bird (i.e., 900,000 rupiah at December 2020 exchange rates) (see Results).

To estimate the number of Javan pied starlings that were traded each year, we used the average number of birds we observed in the 25 bird markets as a more general estimate of their abundance in these markets, and we assumed that these were representative of the at least 56 bird markets that are present on Java and Bali [18]. We then used the turnover estimates to derive the total number of Javan pied starlings that were sold annually in these bird markets.

Data (number of surveys for each bird market, number of birds recorded in each bird market, the mean number of birds recorded in each bird market, asking prices) were log-transformed prior to analysis to approach a normal distribution more closely. We compared the mean number of birds we recorded in the 15 more intensely monitored markets with the means of all 25 markets using a t-test; we tested for correlations between market size, the number of surveys that were conducted, the total number of Javan pied starlings that were recorded, and the mean number of birds recorded in each market by calculating Pearson's correlation coefficients. We compared asking prices in the 1980s, 2000s, and the 2010s (no data were available for the 1990s) with a one-way ANOVA followed by post hoc Tukey's tests. All statistical tests were two-tailed with significance accepted when $p < 0.05$. We present means $\pm$ one standard error of the mean (s.e.m.).

### 3. Results

*3.1. Overview of Trade in Javan Pied Starlings*

In all, we recorded 24,358 pied starlings. The percentage of *jalla* type birds in the individual assessments ranged from 84.6–100% with a mean of 96.3 $\pm$ 1.3. Overall, 8.1% of the birds were of non-*jalla* type, but this was influenced by a relatively large number of these types having been observed in Pramuka, a known center for the import of songbirds from other parts of Southeast Asia. Taking the higher estimate of 8.1% of these being non-*jalla* type birds, this suggested that 22,380 were Javan pied starlings.

The mean number of birds per survey in the 25 bird markets was 70.4 $\pm$ 25.0 ($N = 25$), whereas in the 15 more intensely monitored markets it was 103.0 $\pm$ 39.6 ($N = 15$) (Figure 2), and the difference was statistically different (*t*-test, $t = 1.189$, $p = 0.241$). In all but the smallest bird markets, the species was present in just about every monthly survey, and Javan pied starling was thus one of the most common species in trade.

There was no statistically significant relationship between the size of the market and number of surveys (Pearson's R = 0.252, $R^2 = 0.064$, $N = 25$, $p = 0.224$). While there was a positive relationship between the number of surveys we conducted in a bird market and the total number of birds (R = 0.771, $R^2 = 0.594$, $N = 25$, $p < 0.001$), this relationship was no longer significant when the mean number of birds was considered (R = 0.362, $R^2 = 0.131$, $N = 25$, $p = 0.076$). Thus, while the number of visits to individual bird markets differed considerably (from 2 to 38 monthly surveys), markets with more Javan pied starlings on average were not surveyed more often, nor were larger markets surveyed more or less than smaller ones. There was a positive relationship between the size of the bird market and the mean number of Javan pied starlings we encountered (R = 0.779, $R^2 = 0.611$, $N = 25$, $p < 0.001$).

Discussions with commercial breeders in Yogyakarta and nearby Klaten in Central Java, the epicenter of commercial breeding of starlings, revealed that while some breeders did provide some Bali and black winged mynas for releases as part of conservation programs, as required by law [18], this was not the case for Javan pied starlings. While none of the breeders we spoke to were aware of any release projects for this species, on a small scale, birds are being released. For instance, in November 2019, ten were released in Ancol, North Jakarta, by the company PT Pembangunan Jaya Ancol that owns Sea World Ancol and Ocean Dream Samudra. The release was part of the company's aim to restore parts of the former birdlife on their grounds as, according to its Vice-President, Java pied starlings

had disappeared in the aftermath of several waves of avian influenza that hit Indonesia in the mid-2000s [21].

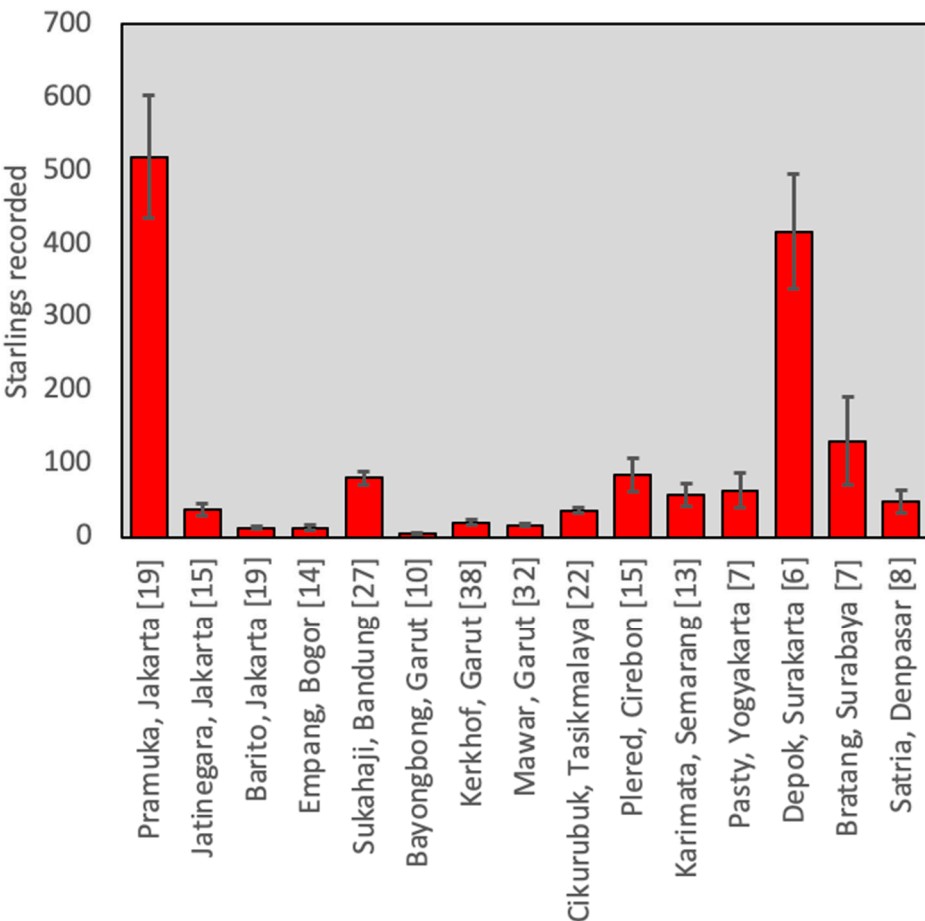

**Figure 2.** Javan pied starlings for sale in 15 bird markets in 11 cities on Java and Bali, Indonesia (cities are listed from west to east), showing the mean ± s.e.m. of the number of birds observed per survey (the number of surveys for each market are indicated in square brackets).

### 3.2. Monetary Value and Total Numbers in Trade

While the Javan pied starling was a cheap songster in the 1980s, when the birds were all sourced from the wild, it appeared that asking prices were up substantially in the early 2000s (no price data is available from the 1990s). By the early 2000s the birds were bred in considerable numbers, and wild birds commanded premium prices. For instance, in 2007, an adult captive-bred bird could be bought for USD23, whereas the price for a wild-caught adult ranged between USD112–157. In 2011 a captive-bred immature was USD35, whereas a wild-caught immature was USD51–61. Mean prices stabilized in the 2010s at ~USD65, and mostly (or even exclusively) captive-bred birds were offered for sale. Prices in the 1980s, 2000s, and 2010s (including 2020) differed significantly (one-way ANOVA, $F_{2117} = 74.049$, $p < 0.001$), with the differences between 1980s and 2000s and 1980s and 2010s being significant (post hoc Tukey's test, $Q = 14.79$, $p < 0.001$ and $Q = 15.10$, $p < 0.001$), but not the difference between the 2000s and 2010s ($Q = 0.31$, $p = 0.974$) (Figure 3).

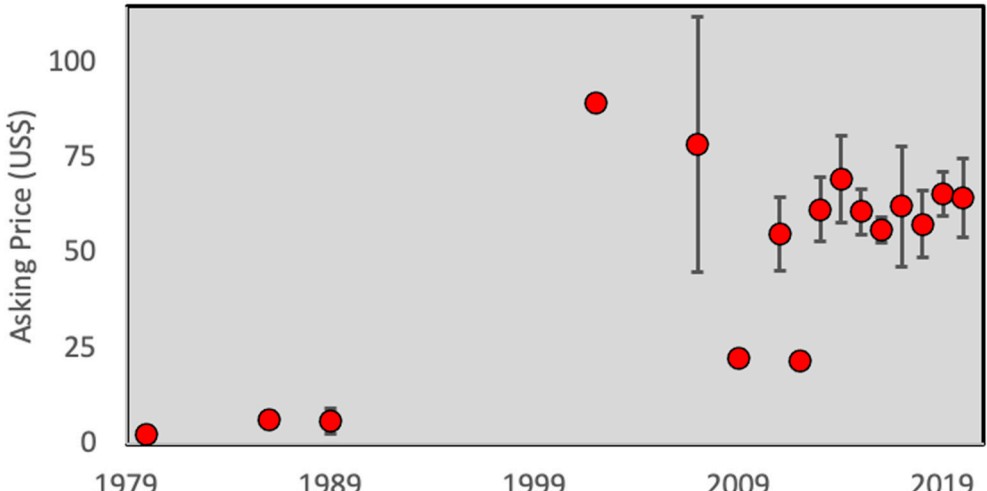

**Figure 3.** Asking prices (mean ± s.e.m.; in USD, corrected for inflation to December 2020) for Javan pied starling in Javan bird markets.

We estimated that turnover after five days was 30.6 ± 13.7% and after seven days it was 37.2 ± 6.4% (Figure 4). Using data from the 25 bird markets and extrapolating this to all 56 bird markets in Java and Bali suggested that 76,084 ± 4869 (based on the seven-day turnover numbers) and 85,238 ± 11,678 (based on the five-day turnover numbers) Javan pied starlings are traded each year. This represents a monetary value from USD4.95 ± 0.32 million to USD5.54 ± 0.76 million.

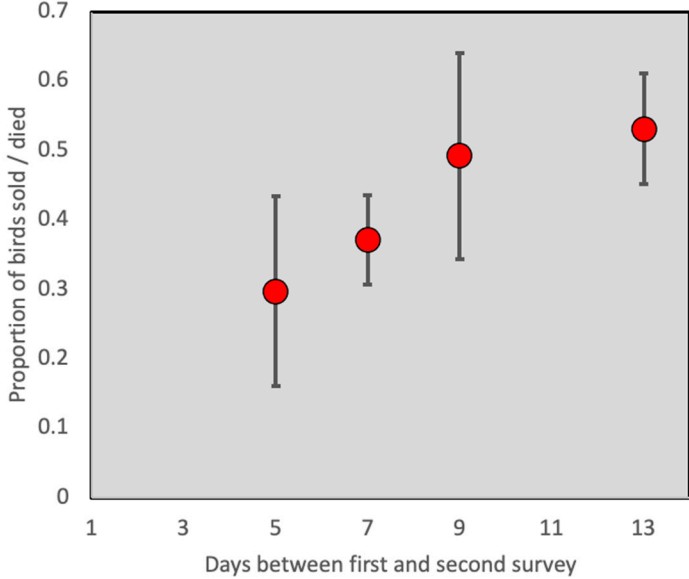

**Figure 4.** Estimates of turnover (proportion of birds sold or died) of Javan pied starlings as a function of the number of days between the first and the second survey, showing the mean ± s.e.m.

## 4. Discussion

### 4.1. Javan Pied Starling Trade and Conservation Discourse

In the 1980s and 1990s, the major conservation NGOs in Indonesia that had bird conservation on Java as part of their remit focused heavily on (montane) forests and forest birds (e.g., [22,23]) or on wetland and coastal bird conservation [24,25]. Open countryside birds and ones that (formerly) were found in urban areas were overlooked. The one project that may have had some relevance to the conservation of Javan pied starlings was that of the Bali myna *Leucopsar rothschildi* in western Bali [16,26], as increased protection of

that species, restoration of its habitat, and increased patrolling most likely would have been beneficial for the Javan pied starling as well. The Indonesian Ministry of Forestry was interested in the conservation of rare and endangered species (often collaborating in these efforts with NGOs) and the utilization of birds (often focusing on harvest quotas and trade) [27]. In various studies on the birds offered for sale in one or more of the bird markets in Java, invariably the Javan pied starling was recorded, often in significant numbers [4,8,28,29], yet it was rare for trade to be flagged up as a serious problem for the species, as the attention was focused on more high-profile birds. The dramatic decline in the number of Javan pied starlings on Java was noted in the late 1990s. However, given that its species-specific status had yet to be recognized, and populations of the species apparently fared well in mainland Asia, again, no specific actions were taken. Illustrative of this is the quote from BirdLife's Threatened Birds of Asia from 2001 [30], where the reasons for the decline of the black-winged myna *Acridotheres melanopterus* are discussed (emphasis added):

> "The precipitous decline in the black-winged starling appears to be attributable to one main factor, human exploitation as a cagebird. Whether, however, other threats, including pesticide contamination and even habitat loss, are also to blame remains open to speculation and in need of study ( . . . ) Java is well known for its cultural attachment to birds in captivity. The almost total disappearance of the black-winged starling from most parts of its range is apparently an example of the power that relentless live animal trade can exert, and which has turned all open and semi-open landscapes on Java into the eerily birdless spaces for which the island has become so notorious ("nothing flies across the road, soars overhead or hops about in a hotel garden") ( . . . ) The worst affected by this dominant cultural interest in caged wildlife, in global terms, are inevitably those animals with the best qualities (looks, voices, abilities) and the smallest ranges, and a handsome and distinctive bird such as this, confined to lowland areas in Java, Bali and Lombok, was predictably (but not, alas, predicted to be) a candidate for serious decline. In fact, amongst the starlings, the Asian pied *Sturnus contra* has fared far worse on Java than the black-winged ( . . . ), but mercifully has a much wider range."

Every part of this quote has turned out to be as true for the Javan pied starling as it is for the black-winged myna. The conservation discourse for the black-winged myna, however, took a very different direction than that of the Javan pied starling. For the former, there was a strong focus on legislative protection, bird market surveys, captive-breeding by zoos and conservation organizations (in addition to commercial captive breeding to meet the demand for cagebirds), and ultimately releases of birds back into the wild [5–7,18,31], whereas for the latter, no conservation actions were initiated [14]. The only area of research in Javan pied starlings that showed strong growth since around 2010 is that of the business of breeding and selling, and this research was spearheaded by researchers from economy, business, and agricultural department in several Indonesian universities [32–34]. The discussions about the Javan pied starling that are held online, on specific cage bird lovers' forums such as OmKicau, and in Indonesian magazines all have a focus on commercial breeding, selling and buying of birds, the training birds should receive to reach their maximum potential as a songster, and, indeed, there is a large online presence of commercial traders that participate in this debate. Conservation or preservation of wild birds rarely, if ever, makes an appearance in these discussions. Only in recent years, as part of the Asian Songbird Crisis, the has attention shifted, and now trade is indeed recognized as a serious impediment to the conservation of Javan pied starlings [1,6]. Thus far, this attention is fronted by international organizations, zoos in Europe and the USA, international research teams, and less so by Indonesian organizations.

### 4.2. Javan Pied Starling Numbers in Trade

It is unclear what the longevity is of a Javan pied starling in captivity, but on Javan pied starling online forums and discussion groups, an age of ten years is seen as advanced. If we assume that a Javan pied starling bought in a bird market can live with its new owner for an average of six years, then our estimate of ~80,000 birds sold each year would suggest that around half a million Javan pied starlings are currently in captivity on Java and Bali. It is unclear how many are sold in venues other than bird markets (Figure 5), but direct supplier to customer and online retail would add substantially to this total. Based on a household survey conducted in 2006, Jepson and Ladle [35] estimated the captive population in six cities on Java and Bali (Jakarta, Bandung, Yogyakarta, Semarang, Surabaya, Denpasar), at 13,626–26,832 birds. These six cities represent ~12% of the population on these islands (or ~20% of the urban population), suggesting that in the mid-2000s, the total number of Javan pied starlings in captivity on Java and Bali was in the order of one to two hundred thousand birds. More recent estimates, also based on household surveys, suggest that the number of Javan pied starlings in all of Indonesia is considerably higher, at $1.14 \pm 0.14$ million birds [12].

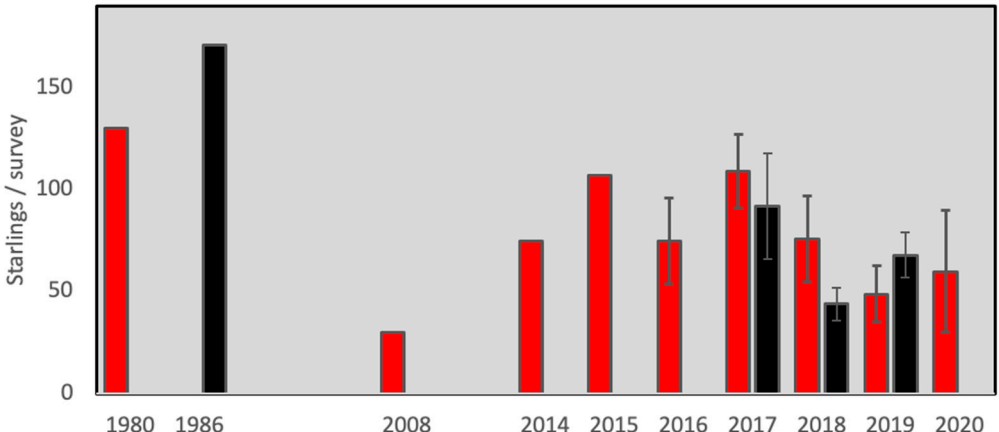

**Figure 5.** Number of Javan pied starlings (mean $\pm$ s.e.m.) offered for sale at two bird markets on Java (red, Sukahaji, Bandung West Java; blue, Karimata, Semarang, Central Java) showing consistent availability of the species (from [9,28,29,36] and this study).

The discourse of the trade and demise of the Javan pied starling in the wild can be broken down into three distinct periods. In the first period, running up until probably around the mid-1990s, there was a strong link between the wild and captive population, as most, if not all, birds in captivity were derived from the wild. Javan pied starlings were also cheap. While in the later years of this first period there may have been a perception that the species was still common in the wild, in reality, only small numbers were recorded [14]. In the mid-1990s and possibly up until the mid-2000 there was an arrival of a substantial number of captive-bred birds onto the market. Javan pied starlings were still captured from the wild and in trade, a premium was paid for wild-caught birds. The latter part of this second period was also the time when the first commercial breeding facilities were set up and although hard data are lacking, we find it plausible that the establishment of these facilities, their need for a genetically varied parent stock, and the access to financial means may have increased the pressure on the last remaining wild birds. While a premium of wild-caught birds may have resulted in these birds being less affordable for the average hobbyist, this was not an impediment for commercial companies. This period was also the last from which we have reliable records of Javan pied starlings in the wild [14]. The third period, from the mid-2000, saw an increase in the commercialization of the breeding and the trade in Javan pied starlings and the emergence of a disconnect between wild birds (if any were still remaining) and captive ones. We expect that for at least part of the trade in Javan pied starlings, this disconnect will only grow in the years to come. We

furthermore anticipate that some breeders will focus on producing "designer starlings" by selective breeding, cross-breeding Javan pied starlings with other species of starlings (including other Asian pied starlings, black-winged mynas, and Javan mynas *A. javanicus*), and breeding leucistic or piebald birds.

### 4.3. Potential for Reintroduction or Population Re-Enforcement

Hitherto reintroduction or re-enforcement (i.e., the intentional movement and release of an organism into an existing population to boost the populations) of various starling species have been undertaken on Java and Bali. In the 1980s the first captive-bred Bali mynas were released into Bali Barat National Park [26]. Later, the species was reintroduced in central Bali [37], and a population of released birds has established itself on the islands of Nusa Penida and Nusa Lembongan off Bali [16,37,38]. In 2012 and 2013, the first captive-bred black-winged mynas were released in West Java and Banten [5], and this was followed by several other releases in western Java. One of the many challenges with these release projects is the limited number of birds that are available for release at any one time. Thus, for both species, typically 5 or 10 and up to 40 captive-bred birds were released in any given year. The situation with Javan pied starlings is different, as there is no shortage of captive-bred birds to be released and certainly, when purchased in bulk, prices are not excessive. We estimate that a thousand captive-bred Javan pied starlings of slightly different ages could be purchased in a single season from commercial breeders for around USD25,000. It is tempting to try to recreate the past—in the late 1900s in the cities of Brebes and Tegal along Central Java's north coast, starlings would gather in the thousands to roost [39]—by releasing large numbers, hundreds or even thousands, of Javan pied starlings at single sites. We would warn against this until (a) it is better understood what the driving forces were between the rapid decline of the species and (b) a situation is created whereby the poaching of songbirds has been curtailed. The one release that we document for the Javan pied starling by a private company on its own grounds was small-scale (ten birds), and the reason that was given for the species disappearance (bird flu) [34] may indicate that limited attention was given in this project to the potential risks of poaching.

For a brief moment, in the months of July and August 2018, the Javan pied starling was included in Indonesia's protected species list, as it was recognized that the last remaining wild individuals of the species were indeed in need of legal protection. Under pressure from commercial Javan pied starling breeders, hobbyists, and bird traders, this decision was reversed. The large number of Javan pied starlings in captivity apparently precluded the need for the legal protection of the species in the wild. For at least the last decade, although the species is not legally protected, it was not allowed to harvest wild birds for commercial purposes, as no harvest quota had been allocated for the species. If any re-enforcement of reintroduction program is to be initiated, then an increase in legal protection of the species or the specific population may be warranted. While this may not be possible at a national level at present, it could still be achieved at a provincial or district level.

### 4.4. Role of Zoos in the Conservation of Javan Pied Starling

Since 2015, Indonesian NGOs and European zoos have an established co-operation through the Threatened Asian Songbird Alliance (TASA), aimed at addressing the Asian Songbird Crisis through, among other activities, the establishment of conservation breeding programs for several bird species [1]. Successful precedent cases of such kinds of programs already exist, such as the working relationship between the Cikananga Wildlife Center in Java and several European zoos to breed and recover populations of highly endangered Indonesian passerine birds, including the black-winged starling [5]. In addition, in 2016 the IUCN Species Survival Commission established the IUCN/SSC Asian Songbird Trade Specialist Group with the aim of preventing the imminent extinction of songbirds threatened by unsustainable trapping and trade in wild-caught songbirds. As of April 2021, the Zoological Information Management System (ZIMS) contained data on 30 Javan pied starlings in two accredited zoos (Jurong bird park, Singapore and Taman Safari Pasuruan,

Indonesia), but others are present in zoos in Java, such as Taman Mini Indonesia Indah's Bird Park, that have no links to ZIMS. However, given the very large numbers of Javan pied starlings in captivity in Java and Bali, in our view, there is no urgent need for zoos to get involved in any captive breeding program of this species.

Zoos should instead partner with commercial facilities and local and international NGOs to offer capacity building and financial and technical support in educating people to support captive-breeding programs intended to help conservation. For instance, zoos can give support to and conduct community outreach and awareness campaigns on the current worrying conservation status of songbirds in Indonesia and stimulate positive changes in society towards protecting these species. One of such important campaigns was already started in 2017 by the European Association of Zoos and Aquaria (EAZA)'s Silent Forest Group, which has been promoting environmental education activities in-region in Southeast Asia and at European zoos, using six Asian songbird species as flagships [40]. We argue that the Javan pied starling shows great potential to be used as a flagship species in educational initiatives due to its critical conservation status and its long historical use by and high appeal to Indonesian residents as a prized domestic cagebird. In addition, EAZA has produced several guidelines for the management of animal species in captivity, including specific bird species (e.g., [41]). The ones that are available in Indonesia for the Javan pied starling are written for a commercial market and focus mainly on keeping costs low while maximizing output. Zoos could build on these manuals to develop more appropriate husbandry and best practice guidelines for the Javan pied starling and other Asian songbird species and offer technical support for its application in these breeding centers and NGOs.

Finally, the Asian Species Action Partnership from IUCN has recognized the need for surveys of the remaining wild bird populations and conducting research on endangered Asian species. Therefore, zoos could also participate in the discussions for setting research priorities to be undertaken to gain a broader picture of the current status of the Javan pied starling and promote fundraising campaigns to offer financial support to research activities. Planning a reintroduction program for the species following release protocols approved by international bodies first requires proper research on the species' in situ status, genetic load, and original range distribution, along with good understanding of the events behind the extinction of the Javan pied starling in nature. The first steps in this process have already been taken [13,14], and in east Java, the Prigen Conservation Breeding Ark, a collaborative breeding project in which, amongst others, Taman Safari Indonesia participates, has been established with the aim of setting up an insurance population for future release.

Zoos can also play an essential role by cooperating with commercial breeders that take pride in producing "pure" Javan pied starlings and that can trace their stock back to wild-caught birds on Java or Bali. As long as the in situ information required for an efficient reintroduction program is achieved and poaching is efficiently curbed, at some point, these "pure" individuals could be used to establish viable populations of the Javan pied starlings in the wild.

## 5. Conclusions

We show a unique case of a species that became extinct in the wild seemingly without researchers and conservation practitioners noticing it. We report, a posteriori, on a substantial trade in Javan pied starlings between 2014 and 2020 on the islands of Java and Bali. The number of birds that we estimated that are sold in the bird markets, ~80,000 each year, are in stark contrast with the number of birds that are remaining in the wild, which is close to zero. Asking prices for Javan pied starlings that have been bred commercially have stabilized in recent years, suggesting that supply can keep up with demand. The Javan pied starling may be a species that, given strict protection, might rapidly recover its numbers, most likely in a few selected well-managed areas. A carefully planned formal release scheme, whereby a number of key stakeholders, including commercial breeders and zoos, participate, could be a part of this recovery program. It is too early to give

clear directions what should be included in such a recovery plan, but it is evident that careful planning is needed to determine the most appropriate further course of action. Furthermore, the case we present is worrying and suggests that the threat for this species, as well as other bird species, is not under control. Reintroductions can be a measure to restore extirpated populations, but the threat may persist if adequate actions to reduce the threat are not taken.

**Author Contributions:** Conceptualization, V.N. and K.A.I.N. methodology, V.N., M.C., H.R.E.B., T.Q.M.; formal analysis, V.N.; investigation, V.N., A.A., T.D., K.H., R.H., A.L.; data curation, K.A.I.N., K.H., T.D.; writing—original draft preparation, V.N., M.C.; writing—review and editing, K.A.I.N., H.R.E.B., T.Q.M., K.H., M.B., A.V.W.; supervision, K.A.I.N., B.B., M.C., M.A.I.; project administration, K.A.I.N.; funding acquisition, K.A.I.N., V.N. All authors have read and agreed to the published version of the manuscript.

**Funding:** This research was funded by Cleveland Zoo and Zoo Society, Columbus Zoo and Aquarium, Disney Worldwide Conservation Fund, Global Challenges Fund, Henry Doorly Zoo, Lee Richardson Zoo, Little Fireface Project, Mohamed bin al Zayed Species Conservation Fund (152511813), Moody Gardens Zoo, Naturzoo Rhein, Paradise Wildlife Park, People's Trust for Endangered Species, Sacramento Zoo, Shaldon Wildlife Trust, and ZGAP. TQM is funded by Wildlife Conservation Society, Wildlife Conservation Network, and British Federation of Women Graduates.

**Institutional Review Board Statement:** This study did not involve any experiment on animals or research involving human participants. Informal discussions with breeders followed the ethical guidelines proposed by the Association of Social Anthropologists of the UK and Commonwealth.

**Informed Consent Statement:** Not applicable.

**Data Availability Statement:** The data presented in this study are available on request from the corresponding authors.

**Acknowledgments:** Parts of this research are underpinned by a Memorandum of Understanding between Oxford Brookes University and Universitas Gajah Mada, and we thank the respective deans for their support. We thank the Indonesian Institute for Sciences (LIPI) and the Ministry of Research and Technology of the Republic of Indonesia (Kemenristek) for authorization. We thank the three reviewers for constructive comments and feedback.

**Conflicts of Interest:** The authors declare no conflict of interests. The funders had no role in the design of the study; in the collection, analyses, or interpretation of data; in the writing of the manuscript, or in the decision to publish the results.

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
