# Peer review of "Large-Scale Trade in a Songbird That Is Extinct in the Wild"

_diversity, doi:10.3390/d13060238_

Round 1

Reviewer 1 Report

A very interesting and worth publishing article on a very important problem. Before publishing, however, Authors should clarify a few points. The main problem of the work is the poor description of the methods of data collection and analysis. The problem presented in this paper is difficult. Not all data is available in such studies, and the markets themselves vary over time. Nevertheless, certain data collection frameworks should be better described in order to understand the results obtained.

The most important comments are below:

Title - I would consider including in the title the name of the taxon analyzed in the article. This would raise interest in the problem of conservation of this species

L25 - ".. with ~4% 25 showing signs of introgression with other taxa." I do not see this information in the article. The abstract should not contain new results, not presented in the article.

L87-88 - redundant information

L97 - "2 to 38" - What did the number of surveys in each of the markets depend on?

L98-99 -"cross checking the ability to identify animals" - Could you clarify?

L101-103 - I do not fully understand the principles of data collection. Only observation, or also talks with sellers. How visible were all the birds at the analyzed fairs (available for counting by the person performing the counting)? Could there be birds hidden somewhere that the sellers did not reveal. From the estimate turnover way, do I understand that the salespeople did not cooperate or were not asked by the interviewers?  Please specify the data collection rules.

L115-117 - Very unclear fragment. I don't understand how to estimate turnover. Especially after reading the information in L167-169. Clarify

L122  enter full names for the first time when entering abbreviations. Only then use abbreviations.

L85 - 2.1. Data acquisition section

Can you divide this section into at least two parts, detailing your bird market data collection and price information from the information you start on the L 122 line.  What is the result of the data set and what is the result of interpredation or various assumptions for the analysis of results? "

L129 - What do you mean by "We adopted this approach" from [16, 17]. Have you made some changes or have you applied exactly? Clarify

L137-147 - The sources and methods of collecting information are too imprecise. In my opinion, the methodology should be described in such a way that we can repeat such research (taking into account, of course, the difficulties associated with this type of research). Some details not included may therefore have an effect on the results.

L 150 "We used data from all 25 bird markets to establish the relationship between bird market size .... Table 1) but in table 1, n = 16. Why this difference?

L155-166. I suggest moving the bird calculations to the result section

L167-169. I see some discrepancy between these lines and L114-117. Crarify. Generally, you should better divide information about data collection and the principles of data analysis. Currently, information is mixed in both chapters.

L171-176. Small consignment of birds may have been undetectable. If so, it should be noted in the manuscript. There is a concern here that the data on birds sold or dead are understated in the manuscript. Such a situation may arise when the deliveries were small and only partially supplemented by the birds sold. Such deliveries did not result in an increase in the number of birds at the market compared to the previous survey

L194 expand the abbreviation s.e.m .. when first used in the text ..

L199-200 - These are means calculated on the basis of 25 and 15 means, or means based on individual surveys. In the latter case, s.e is very high

L201 - Test. - What was the statistic unit, bird markets or surveys?

L204-212 - Please, apart from R2 and p value, give N - the number of analyzed cases. I have some doubts how many cases you analyze, which may affect the statistical significance of the relationship. It's easier to get it with more cases

Reviewer 2 Report

Please see attached word document

Reviewer 3 Report

Minor spelling edits:

Line 53 - Baveja et al (and not Beveja)

Line 305 - Jurong (and not Jurung)

Just a few minor queries/comments that the authors could consider addressing in their paper:

  • The paper briefly mentioned songbird competition(s) - it would be good to elaborate a little bit more as to whether the species is also prized in competitions beyond just appealing to bird hobbyists/keepers as a household companion
  • The study indicated that there was a period where wild-sourced birds were priced higher than captive-bred individuals (this was when captive-breeding was just starting off) - for some other songbird species, captive-bred birds were considered more costly due to the resources invested in breeding but also training etc., and it was cheaper to catch wild birds. It is interesting that this is the reverse for the JPS. Did the authors enquire with breeders as to whether the higher costs were perhaps also associated with wild-sourced birds possessing better physiological traits (e.g. better singing capacity)?
  • Conservation recommendations made by the authors included legal protection as well as a breeding for release conservation program. How about habitat restoration and/or protection - how essential is this layer? Java/Bali have been extensively cleared/modified, and apart of removing the threat of poaching or understanding the drivers of decline, it is curious as to whether there are any appropriate and/or protected habitats left for future bird releases. This is key as the IUCN SSC recommends a more holistic one-plan approach to species conservation (i.e. in situ and ex situ components) 
  • There is a conservation authority (i.e. the IUCN SSC Asian songbird specialist group), as well as a conservation strategy - it would be good to make some reference to these, especially in supporting some of the actions proposed in the paper. And perhaps the paper can also more strongly recommend the endorsement of the development of such a conservation plan/program by this authority
  • The paper mentioned that other species of pied starlings were observed/imported and that there is a potential rise of a market around "designer starlings" - just curious if the authors enquired with songbird keepers/breeders on the levels of demand for non-native species, and if so which species/from where, is there a preference? Do the authors also think that hybridisation with non-natives is increasingly (or will be) becoming a threat? As this has potential negative spill-over effects on wild populations (i.e. release of morphologically JPS-birds but actually hybrids, or non-native escapees resulting in hybridisation in the wild)

Overall, this paper was well written, timely, but also addresses key and unique topics around the conservation of a highly threatened but yet "common" bird species 

Round 2

Reviewer 1 Report

The authors made sufficient corrections to publish the manuscript. Interesting article. Congratulations.

Reviewer 2 Report

I thank the authors for their careful attention to my concerns, the manuscript reads well and will be an important contribution.